# EMRI_MC: A GPU-based code
# for Bayesian inference of EMRI waveforms

**Ippocratis D. Saltas**$^{a\S}$, **Roberto Oliveri**$^{b\dagger}$

$^a$ *CEICO, Institute of Physics, Czech Academy of Sciences*
*Na Slovance 2, 182 21 Praha 8, Czech Republic*

$^b$ *LUTH, Laboratoire Univers et Théories, Observatoire de Paris*
*CNRS, Université PSL, Université Paris Cité,*
*5 place Jules Janssen, 92190 Meudon, France*

## 1 Summary

We describe a simple and efficient Python code to perform Bayesian forecasting for gravitational waves (GW) produced by Extreme-Mass-Ratio-Inspiral systems (EMRIs). The code runs on GPUs for an efficient parallelised computation of thousands of waveforms and sampling of the posterior through a Markov-Chain-Monte-Carlo (MCMC) algorithm. EMRI_MC generates EMRI waveforms based on the so–called kludge scheme, and propagates it to the observer accounting for cosmological effects in the observed waveform due to modified gravity/dark energy. Extending the code to more accurate schemes for the generation of the waveform is straightforward. Despite the known limitations of the kludge formalism, we believe that the code can provide a helpful resource for the community working on forecasts for interferometry missions in the milli-Hz scale, predominantly, the satellite-mission LISA.

Code available at https://doi.org/10.5281/zenodo.10204186.

## 2 Statement of need

In view of the burst of GW astronomy and the need for parameter estimation [1–4], including possible effects of modified gravity, EMRI_MC provides a simple, yet efficient code for the GW community of astrophysics and cosmology towards parameter estimation and forecasts for the future LISA detector [2, 5–7].

Parameter forecasting for EMRI signals is not an easy task, because of the challenge to model their waveforms and the high-dimensional parameter space that needs be explored. Most of the attempts in the literature to date are based on the kludge scheme for the waveform generation, as well as on the Fisher information matrix approach for the parameter

---

$^\S$email: saltas@fzu.cz
$^\dagger$email: roberto.oliveri@obspm.fr

forecast; see, e.g., [8–13]. An early attempt at parameter estimation using Bayesian inference was performed in [14] and, more recently, in [15, 16].

`EMRI_MC` relies on four main elements: **i)** the waveform generator; **ii)** the inclusion of the amplitude damping and modified speed of GWs; **iii)** the posterior sampling through MCMC methods; **iv)** the GPU-based vectorisation of quantities such as the likelihood, in order to accelerate computations. Our code aims to provide a simple and efficient tool that could be of help for the community working on the interface of GWs and modified gravity. We emphasise that the structure of the code provides for enough flexibility to allow extension and improvements in each of its elements, according to the need of the specific task.

  **i)** *The waveform generator.*
  Our choice for the waveform generator relies on the popular *Analytic Kludge* (AK) model for the generation of inspiraling EMRI waveforms [8, 9]. Though it is not the most accurate waveform model to date, this choice is justified as follows. AK waveforms provide a sufficiently good approximation of the binary's dynamics, as long as one remains sufficiently far away from the merger. Additionally, AK waveforms allows for an analytic handle on the physics. The equations can be consistently extended with new post-Newtonian and self-force corrections, as well as the inclusion of new physics such as dark matter effects. In this regard, they provide an excellent proxy to perform parameter estimation for future missions, as well as investigate the significance of effects related to new physics.

  The AK model can be replaced with a more accurate waveform model generator. Examples are the *Augmented Analytic Kludge* [11, 17], the *Numerical Kludge* [18–21], the *Fast EMRI Waveforms* [22, 23], and the *Effective One-Body* approach [24–27].

 **ii)** *The inclusion of modified gravity effects.*
  We include effects beyond General Relativity during the propagation of GWs on the cosmological background. Specifically, we include the effects of damping of the amplitude and modification of the GW speed [28–30].

**iii)** *The posterior sampling.*
  Most of the codes currently available for posterior samplings for EMRIs parameter space make use of the Fisher information matrix. We adopt a Bayesian approach using Markov-Chain-Monte-Carlo (MCMC) methods. These are implemented via the MCMC package `emcee` [31], which employs an affine-invariant ensemble sampler [32].

 **iv)** *The GPU-based vectorisation and parallelisation features.*
  Bayesian inferences through MCMC methods are rather expensive for CPUs, especially when posteriors evaluations involve high-dimensional parameter space. To overcome this computational limitation, the code adopts GPU-based vectorisation and parallelisation features.

# 3  Theoretical background

**Waveform generation:** The generation of accurate GW waveforms for binary systems and efficient parameter estimation is key for current and future GW missions such as LISA. For

EMRIs, the accuracy of the waveform in the inspiral phase requires adiabatic, post-adiabatic and self-force approximations [22, 23, 33–39].

The AK model we adopt in our code relies on the Peters-Mathews formalism [40, 41], where adiabatic and post-Newtonian approximations are adopted. The main advantage thereof, for our purposes, is the analytic command over the waveform generation and its parameter space. AK model can also be easily extended and modified in the presence of new physics. For technical details on the theoretical framework and equations we will be using, we refer to [8] and references therein.

The system of equations consists of two main parts: **i)** the equations describing the orbital dynamics of the small body with mass $\mu$ around the central black hole with mass $M$ and spin $S/M^2$, and **ii)** the equations for the generation of the waveform under the quadrupole approximation. The first ones form a system of ordinary differential equations (ODEs) as

$$\frac{d\mathbf{Y}}{dt} = f(\mathbf{Y}(t); \boldsymbol{\theta}), \tag{1}$$

where the vector $\mathbf{Y}$ denotes the orbital parameters $\mathbf{Y} = \{\Phi, \nu, e, \gamma, \alpha\}$, i.e., the phase ($\Phi$), the orbital frequency ($\nu$), the eccentricity ($e$) and two precession angles ($\gamma, \alpha$). The vector $\boldsymbol{\theta}$ denotes the free parameters in our waveform generation model $\boldsymbol{\theta} = \{M, \mu, S, \ldots\}$, i.e., the masses, spin, angles, parameters due to propagation, etc. We note that we work in **cgs units**. For example, we restore powers of $G$ and $c$, define the post-Newtonian order parameter $x = 2\pi\nu GM/c^3$, and the spin magnitude of the central black hole as $0 \leq S/M^2 \leq 1$.

The solution of the orbital equations under the quadrupole approximation allows for the computation of the waveform as

$$h_{ij}(t) = \sum_{n=1}^{n_{max}} h_{ij}^n(t) = \sum_{n=1}^{n_{max}} A_{(n)}^+(t, \boldsymbol{\theta})e_{ij}^+(t) + A_{(n)}^\times(t, \boldsymbol{\theta})e_{ij}^\times(t), \tag{2}$$

where it is understood that $A = A[\mathbf{Y}(t), \boldsymbol{\theta}]$. The polarisation coefficients are computed under a harmonic decomposition up to some overtone $n_{\max}$, according to [40]. We notice that the detector response function, assumed to be included in the above expression, introduces three extra angles on top of the ones related to the orbital dynamics of the system.

**Waveform propagation:** Assuming a plane GW travelling far away from the source through the cosmological medium, we can write Eq. (2) as $h_{ij}(t) = h(t)e_{ij}$, and expand the amplitude of each mode of the wave in Fourier modes with spatial wavenumber $k$ as

$$\ddot{h} + 3H(\tau)(2 + \alpha_{\mathrm{M}})\dot{h} + k^2(1 + \alpha_{\mathrm{T}})h = 0. \tag{3}$$

$H(\tau)$ is the Hubble parameter, $\tau$ the cosmological time, and the quantities $\alpha_{\mathrm{M}}, \alpha_{\mathrm{T}}$ parameterise effects beyond General Relativity modifying the friction and the wave's propagation speed respectively [28–30, 42]. In redshift domain, and under the WKB approximation, one can solve analytically Eq. (3) to find [29]

$$h(z) = h_{\mathrm{MG}} \times h_{\mathrm{GR}} \equiv \frac{1}{\Xi} \times e^{-ik\Delta T} \times h_{\mathrm{GR}}, \tag{4}$$

with

$$\Xi(z) \equiv \frac{d^{\mathrm{GW}}(z)}{d^{\mathrm{EM}}(z)} \exp\left(\frac{1}{2}\int_0^z d\tilde{z}\frac{\alpha_{\mathrm{M}}(\tilde{z})}{1+\tilde{z}}\right), \quad \Delta T \equiv \exp\left(-ik\int_0^z d\tilde{z}\frac{\alpha_{\mathrm{T}}(\tilde{z})}{1+\tilde{z}}\right), \quad (5)$$

$z$ the redshift to the source, and $h_{\mathrm{GR}}$ the contribution one gets from solving Eq. (3) for $\alpha_{\mathrm{M}} = 0 = \alpha_{\mathrm{T}}$. The possible cosmological evolution of $\alpha_{\mathrm{M}}(z), \alpha_{\mathrm{T}}(z)$ is model-dependent, however, they are in principle very slowly-varying functions of redshift, tracing the evolution of the dark energy density fraction. For a discussion on parametrisations of their time dependence we refer to [43]. For the sake of an example, we choose to parametrise $\Xi(z)$ through the physically well-motivated parametrisation of [44] (see also [45])

$$\Xi(z) = \Xi_0 + \frac{1 - \Xi_0}{(1+z)^n}, \quad (6)$$

with $\Xi_0$ a free parameter. Of course, any other physically-motivated parametrisation is equally good. There are scenarios where the parameters $\alpha_{\mathrm{M}}$ and $\alpha_{\mathrm{T}}$ are also frequency-dependent quantities (see e.g [30] for a detailed exploration) as

$$\alpha_{\mathrm{M}} = F(z, f), \quad \alpha_{\mathrm{T}} = G(z, f), \quad (7)$$

with $F, G$ some well-motivated functions of GW frequency ($f$) and redshift ($z$). An example includes a power series expansion $\sum_n a_n(z)\left(f/f_*\right)^n$. Numerically, such frequency-dependent terms need to act upon a tabulated waveform in frequency space.

# 4 Numerical approach and statistical pipeline

**Overview:** Our goal is to streamline an efficient parameter estimation pipeline to get joint constraints on the free parameters of the model.

As a first step, we define a fiducial model with an associated set of fiducial parameters $\boldsymbol{\theta_0}$, which we use to generate the expected waveform $h[\boldsymbol{\theta_0}](f)$ in Fourier space for this model. This waveform is then used as the (mock) dataset at the heart of our MCMC analysis. At each step of our MCMC, the orbital equations are solved for the current set of parameters $\boldsymbol{\theta}$, and the time-domain waveform computed. The latter is then Fourier-transformed into the frequency domain, yielding the waveform $h[\boldsymbol{\theta}](f)$. It is then compared to the mock dataset via the computation of the log-likelihood, as follows

$$\log\mathcal{L} \propto \frac{1}{2}\frac{\left(h[\boldsymbol{\theta}](f) - h[\boldsymbol{\theta_0}](f)\right)^2}{S_{\mathrm{noise}}(f)}, \quad (8)$$

with $S_{\mathrm{noise}}(f)$ the LISA noise function. We have implemented the noise model presented in [8]. One can easily update it to the more recent LISA noise model [46, 47].

**Waveform computation:** The orbital equations (1) are solved as an initial-value problem on a time grid using standard ODE methods, such as the 7th-order Runge-Kutta scheme. Initial conditions are set at the Last Stable Orbit (LSO) and the equations are integrated backwards for a given time window, typically ranging between few months to one year. The

resolution of the time grid is set at 0.1 Hz, which is the typical choice for LISA. As regards the initial value for eccentricity and angles of the system at LSO we fix these geometrical quantities for both fiducial model and MCMC analysis and vary only masses, spin and other physical parameters. The initial value for the frequency $\nu$ at LSO is set according to [8], $\nu_{\mathrm{LSO}} = c^3/(2\pi GM)((1 - e_{\mathrm{LSO}}^2)/(6 + 2e_{\mathrm{LSO}}))^{3/2}$. For the transition to the frequency domain, we use the method of a Fast Fourier Transform (FFT), under an appropriate normalisation choice.

**Posterior sampling:** The posterior sampling is performed through the Python MCMC package `emcee`, which allows for various sampling techniques and parallelisation features.

**Functionality overview:** Vectorisation is achieved mainly through appropriate use of the `ElementwiseKernel` functionality. The code is optimised for running on GPUs, making use of Python's `cupy` library. Computations such as the waveform and the likelihood are GPU-vectorised reducing significantly the evaluation time. Moreover, we have parallelised the MCMC run by allowing each walker to run in parallel. In overall, for 4 free parameters and 8 walkers, this brings down the evaluation of each MCMC step to about **4.5 seconds**, assuming a waveform integrated over 1 year at resolution of 0.1 Hz.

The code's architecture consists of the following **main files**:

**1. global_parameters.py**: This module defines the values of physical constants in cgs units, the parameters of the fiducial model, geometrical parameters and initial conditions of the binary system, parameters for the ODE solver (e.g., integration time window and grid resolution), and MCMC-related definitions. It also defines the maximum number of orbital overtones `n_max` in the computation of the waveform. A change in the number of the parameters in the MCMC requires adjusting the parameter vector in this module.

**2. waveform.py**: This module defines the set of kludge ODE equations (see Eq. (1)), the waveform generator according to Eq. (2), and some GPU-related functionality. Its main functions reads as follows:

– `eqs()`: Defines the set of kludge ODE equations and returns the right-hand-side of them in the sense of Eq. (1). Notice that in the case of the use of an ODE solver other than the native ones in Python, currently used, the return statement of this function might need to be changed.

– `compute_orbit()`: Computes the solutions of the kludge ODE equations defined in `eqs()`. To solve the system of ODEs, we use the Pythonic framework of `solve_ivp()`. This choice allowed to switch between the different native solvers in this library.

– `waveform()`: It calls `eqs()` and `compute_orbit()`, computes the time-domain waveform including the LISA response function (2) and then performs its FFT, via the function `FFT_gpu()`. The computation of waveforms is implemented fully on cuda. GPU vectorisation and acceleration is implemented with appropriate use of `ElementwiseKernel`. For

computational convenience, the ouputted waveform **does not** include the overall factor of the GW luminosity distance. This is included in the function `iterate_mcmc()` below.

– `compute_fiducial()`: Computes the fiducial model based on the fiducual values defined in *global_parameters.py*. The parameter vector defined in this function needs adjustment when adding/removing parameters in the MCMC run.

**3. mcmc.py:** This module defines the MCMC-related functions and the MCMC iterator. Its main functions reads as follows:

– `lnprior(), lnprob()`: These functions define the log-prior and the log-probability, respectively. They need be adjusted when the set of parameters in the MCMC run is modified.

– `iterate_mcmc()`: It calls `waveform()` to compute the waveform, and the likelihood in frequency domain for a given choice of parameters around the fiducial model using GPU vectorisation.

– `get_noise()`: This defines the LISA noise function for GPU parallelisation through an `ElementwiseKernel`.

– `get_Likelihood()`: This function computes the likelihood according to Eq. (8) in GPU vectorised form through an `ElementwiseKernel`. It is used in `iterate_mcmc()` to compute the likelihood at each MCMC step. Modifications to the GW luminosity distance enter here. This function needs to be adjusted according to any change of parameters in the MCMC run.

**4. propagation.py:** This module defines the functions needed for the propagation of the GW wave through the cosmological background in the presence of any modified gravity effects.

– `get_damping(), get_modified_speed()`: This defines the possible frequency-dependent damping of the waveform's amplitude, or the respective change in its propagation speed, due to modified gravity. It is defined through an `ElementwiseKernel` for an efficient evaluation on the frequency grid. After defining the functional form of the frequency-dependent damping and/or GW speed, one should modify appropriately the computation of the likelihood in the function `iterate_mcmc()`. Detailed comments are provided in the code.

– `dL()`: This function defines the electromagnetic luminosity.

– `dGW_Xi()`: This function defines a redshift-dependent parametrisation of the GW luminosity distance due to modified gravity according to Eq. (6). It should appear as an overall multiplicative factor of the waveform in the likelihood computation in `iterate_mcmc()`. We remind that the function `waveform()` does not include in the outputted waveform the luminosity distance factor.

**5. main.ipynb:** Assuming all parameters and fiducial are properly defined as explained

earlier, this Jupyter notebook calls the main functions to initiate the MCMC run, using the package `emcee`. As a simple choice, we have currently set throughout the numerical computation the source location $\{\theta_S, \phi_S\} = \{\pi/4, 0\}$, the orientation of the spin $\{\theta_K, \phi_K\} = \{\pi/8, 0\}$, $\alpha_{\mathrm{LSO}} = 0$, the angle $\lambda = \pi/6$, the initial eccentricity $e_{\mathrm{LSO}} = 0.3$, and $\gamma_{\mathrm{LSO}} = 0$, $\Phi_{\mathrm{LSO}} = 0$, for the respective initial conditions. These can be straighforwardly modified in the file *waveform.py*.

**Extending the parameters in the MCMC run:** First we notice that, in the vector `p` defining the parameters to be varied in the MCMC, the first three values should be by default the central mass $(M)$, the orbiting mass $(\mu)$, and the spin $(S/M^2)$. These are needed by the ODE solver to solve the equations and are passed as `args = [p[0], p[1], p[2]]`. Therefore, it is advisable to always keep this convention. Now, to add new parameters in the MCMC run one needs to make changes at the following points in the code:

– *global_parameters.py*: Edit the sections on parameters for the MCMC run and parameter values for the fiducial model.

– *mcmc.py*: Edit the vector `p` in the functions `lnprior()` and `lnprob()`. Edit the parameters to be varied in the MCMC in the function `iterate_mcmc()`, including possible modifications in the GW luminosity distance which enters in the computation of the likelihood in `iterate_mcmc()`.

– *main.ipynb*: Extend the vector `p_init_MC` which initialises the walkers with the new parameters. Ensure that values for the initialisation of the walkers is meaningful given the problem at hand, otherwise the MCMC will not converge as expected.

– *propagation.py*: Any new parameters which affect the GW luminosity distance must be also reflected in this module where the definitions of the luminosity distances are placed.

**Computational overhead, ODE solver, overtones summation:** An important computational overhead in the evaluation of each MCMC step comes from the choice of the ODE solver. We currently use the native ODE solvers provided by Python's numerical libraries. However, this can be improved using different, external solvers, such as the ones provided by the *Fast EMRI Waveforms* scheme [22, 23]. We notice that the choice of the ODE solver enters in the functions `waveform()` and `compute_fiducial()`. We should also notice that, the choice of different solvers (e.g. RK23 or LSODA) leads to different computation times. Contrary to the ODE solver, the for-loop which sums over overtones in the function `waveform()`, does not seem to cause any sizable computational overhead for reasonable choices of the maximum overtone $(n_{\mathrm{max}})$, we therefore decided not to implement any vectorisation on it, but we plan to explore this feature further in the future.

# 5  Running the code

Running the code is particularly simple. Placing all files in the same folder, and setting up all parameters as explained above, one starts the notebook *main.ipynb*, and executes the cells. The first cell computes the fiducial model, and the following cells start the MCMC run around the chosen fiducial. The MCMC results are stored in a `.txt` file. Currently, for the sake of an example we consider a 4-parameter case: 3 source parameters (2 masses and 1 spin), and 1 propagation parameter ($\Xi_0$).

  As an illustration, in Figure 1 we plot the computed waveforms for characteristic values of the eccentricity and spin, as computed by the function `plot_waveform()`. The orbital angles have been fixed according to the conventions mentioned earlier. What is more, Figure 2 shows an example corner plot from a MCMC run with 4 free parameters - 3 for the generation (masses+spin) and 1 for the propagation of the waveform respectively ($\Xi_0$). As it can be seen, our constraints on the parameter $\Xi_0$ (see equation (6)), which relates to a modified gravity effect in the propagation of the waveform, are within the same order of magnitude as with very recent results in the literature [16].

# 6  Future improvements

Surely the current implementation of this code can be expanded in different interesting and more accurate ways, for example: **i)** The inclusion of environmental effects in the production of the waveform, such as dark matter or baryonic effects due to accretion of the central black hole. Such effects would introduce amongst other things, a new dissipating channel due to the force of the dynamical friction encountered by the orbiting mass. **ii)** Our current use of the standard kludge equations has been based on a trade-off between simplicity and accuracy, combined with the popularity of this formalism for parameter estimation in the literature. However, its waveforms are known to suffer from certain inaccuracies. Improvement can be achieved by implementing the so–called Augmented Kludge Formalism, or the waveforms of *Fast EMRI Waveforms* discussed earlier, which would need a more involved implementation. **iii)** Finally, the implementation of a more efficient ODE solver for the orbital equations could allow us to achieve even faster iterations in the MCMC sampling run.

# Acknowledgements

We are indebted to Stéphane Ilić for collaboration on an early stage of the project, for his critical advice on MCMC simulations, and for his comments on the final version of the draft. We also thank Leor Barack, George Loukes-Gerakopoulos, Simone Mastrogiovanni, Adam Pound and Nick Stergioulas for discussions. The work of R.O. is supported by the Région Île-de-France within the DIM ACAV+ project SYMONGRAV (Symétries asymptotiques et ondes gravitationnelles). I.D.S. acknowledges funding by the Czech Grant Agency (GAČR) under the grant number 21-16583M. He also acknowledges the warm hospitality of R.O at the Observatory of Paris in November 2022, when the project was initiated. We acknowledge the use of the computer cluster at CEICO/FZU where the code was tested, and warmly

thank its administrator Josef Dvořáček for his helpful input on GPU optimisation, and the use of the "Phoebe" computer cluster.

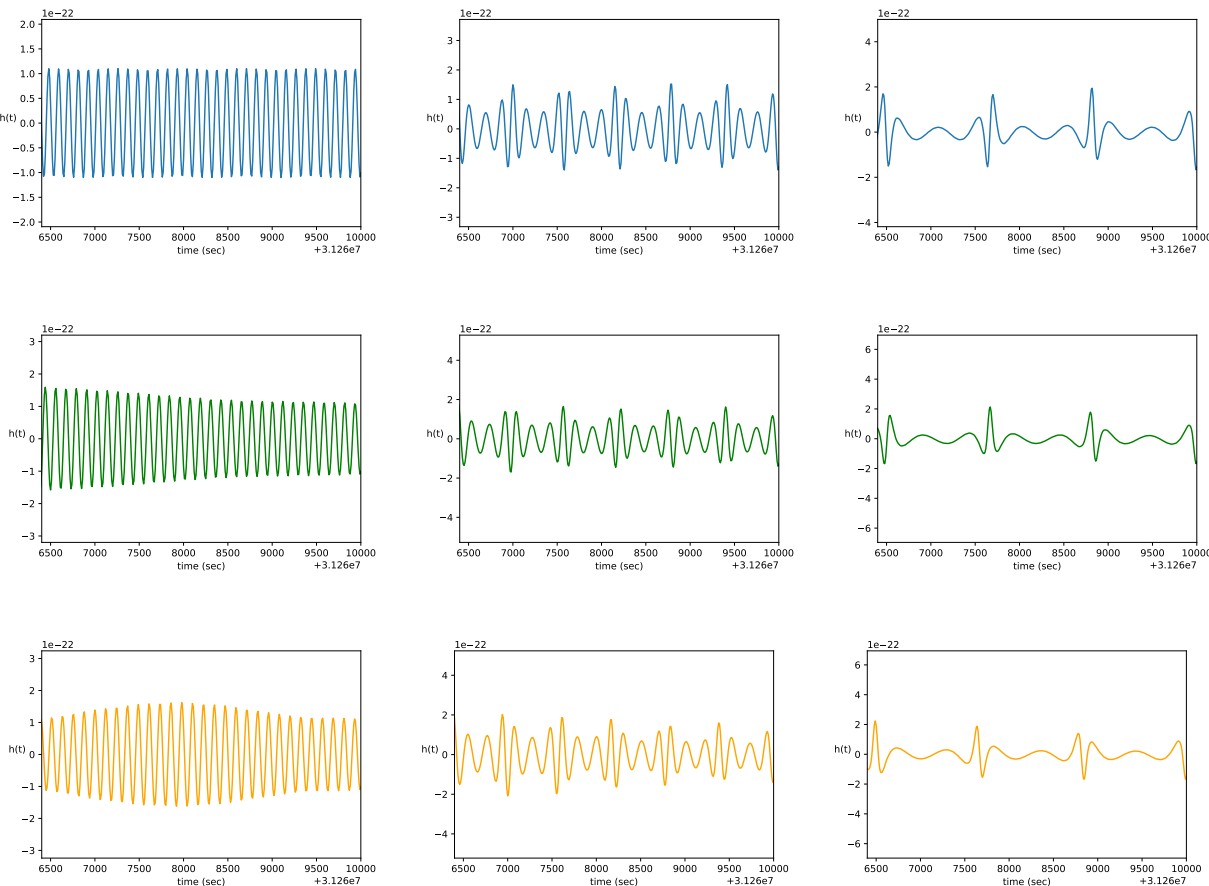

Figure 1: The kludge waveform computed for the last hour before the plunge at the Last Stable Orbit (LSO). Parameters: $M = 10^6 M_\odot$ (central mass), $\mu = 10 M_\odot$ (orbiting mass), $(\theta_S, \phi_S) = (\pi/4, 0)$, $(\theta_K, \phi_K) = (\pi/8, 0)$, $\lambda = \pi/6$ (frame angles), $D = 1$ Gpc (distance to the source). **First row**: $S/M^2 = 0$ (dimensionless spin of the central black hole), $e_{\mathrm{LSO}} = 0, 0.3, 0.6$ (eccentricity). **Second row**: $S/M^2 = 0.4$, $e_{\mathrm{LSO}} = 0, 0.3, 0.6$. **Third row**: $S/M^2 = 0.8$, $e_{\mathrm{LSO}} = 0, 0.3, 0.6$.

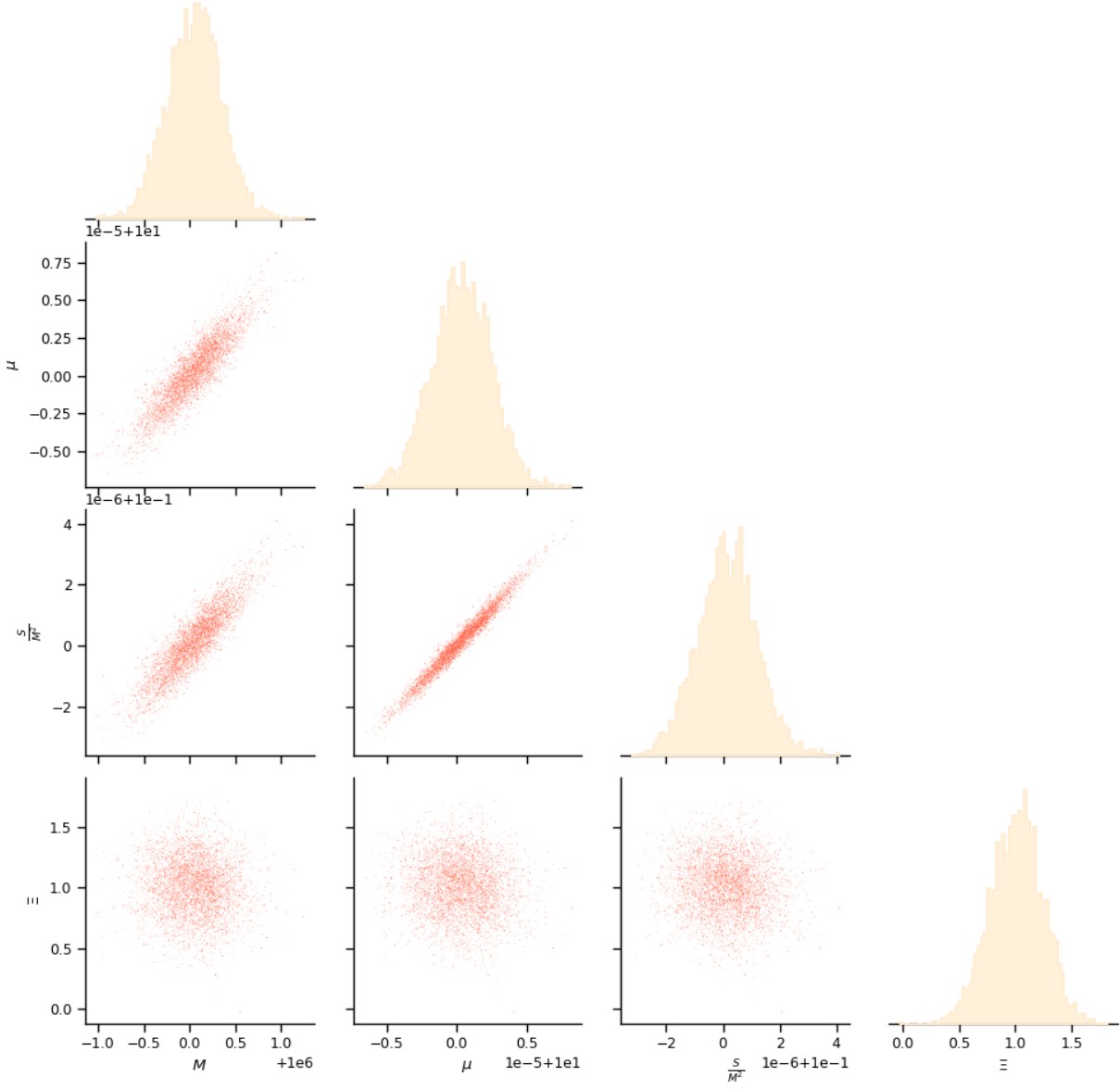

Figure 2: An example corner plot from an MCMC exploration with fiducial/injected values $M = 10^6 M_\odot$ (central mass), $\mu = 10 M_\odot$ (orbiting mass), $S/M^2 = 0.1$ (dimensionless spin of the central black hole), $\Xi_0 = 1$ (no modified GR effects; see Eq. (6)), 2000 steps and 8 walkers. We have assumed an observation of one year. Median and 90% C.I. are $M/(10^6 M_\odot) = 1.28^{+0.148}_{-0.2604}$, $\mu/M_\odot = 10.0000021^{+0.0000010}_{-0.0000019}$, $S/M^2 = 0.1000010^{+0.0000005}_{-0.0000009}$, and $\Xi = 1.1685571^{+0.1043108}_{-0.1965527}$. We have assumed that the distance (redshift) to the source is known, and equal to 1 Gpc. The constraints are somewhat tighter than those in the literature [13], as our MCMC exploration covers a smaller EMRIs' parameter space. The eccentricity and the orbital angles at LSO have been kept fixed in the MCMC run. We have used the LISA noise model of [8].

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
