# Peer review of "EMRI_MC: A GPU-based code for Bayesian inference of EMRI waveforms"

_SciPost Physics Codebases_

## Round 1 · Referee Report · Anonymous (Referee 1) · 2024-8-27

Strengths
1-Detailed, well explained ideas.
2-Implementation of software for accelerated computations for the analysis of Extreme Mass Ratio Inspirals.
3-Waveform models include modified gravity effects.
4-Accompanied with software that is quite easy to use and modify.
Weaknesses
1-Data analysis section needs more detailed descriptions of the methodology that was followed.
Report
Please see attached file
Requested changes
Please see attached file
Recommendation
Publish (easily meets expectations and criteria for this Journal; among top 50%)
Author: Ippocratis Saltas on 2024-10-10 [id 4858]
(in reply to Report 1 on 2024-08-27)Dear Editors and Referee,
Best regards,
Ippocratis Saltas and Roberto Oliveri
A. Comments on the manuscript
1.First general comment: I find that the text is in need of more detailed discussion on the comparison of past implementations of EMRI analyses and the one introduced in this work. I believe that a discussion section (or at least a few paragraphs) should be written, in order to stress the novel ideas introduced here.
2. In section 2, the authors write "Parameter forecasting for EMRI signals is not an easy task, because of the challenge to model their waveforms and the high-dimensional parameter space that needs be explored". This is true, and another challenge is the multimodality of the likelihood, and possible degeneracies (e.g. see Chua & Cutler 2022). Another is the potential overlap and confusion with other signals (transient or stochastic, e.g. see works about the Global Fit of the LISA data).
3. End of section 2, item (iv): Up to this point it is not entirely clear which elements of the analysis are parallelized with GPU hardware. Before section 3, it would be useful to see a short list of items that the authors have improved with GPU parallelisation (e.g. the likelihood and/or the different parts of waveform). Otherwise the reader must go through the complete text, or even the code, in order to read this information.
Section 4: Considering the challenges of the analysis of EMRI signals, I believe that the section 4 is a bit short and lacks details (especially compared to the rest of the manuscript). In particular, it is not clear whether the analysis is performed by using the Time Delay Inteferometry (TDI) channels, or it is done directly in h_c (or similar) units. For the former, a more detailed description is required about the number of channels, and wether a noise orthogonal TDI combination is adopted (this could be assumed from eq. 8, but it should be properly described in the text). If the analysis is done in h_c units it should also be clearly stated and described, because it could introduce simplifying assumptions in the parameter estimation process. Usually this is done by assuming ideal instrument and perfect knowledge of the overall system (for example ignoring transfer function variations for the various noises and signals, as well as other analysis complications such as correlations between channels). A simple plot of some mock data in frequency domain could be useful in this section.
5. Same section: It is not stated wether the analysis is "noiseless", or if the instrumental noise is simulated from the noise curve (or any other method). Maybe it's evident from the code implementation, but I think it should also be clear in the text.
We have clarified this point with a comment after equation (8).
6. Same section on the 'waveform computation' paragraph: Extrinsic parameters are kept fixed at true values for the parameter estimation analysis. This is another simplification in that needs to be stated in the introduction and/or the discussion sections.
7. Same section, 'Functionality overview' paragraph: One can assume that the MCMC walkers are parallelized with multiprocessing (CPU), but it would be better to clearly state it in the text for the readers not familiar with the emcee software.
8. Same paragraph as above: The efficiency and performance of the software is reported. While it's very challenging to directly compare with other software implementations from previous works, I think it would be beneficial to present some ballpark estimates for comparison.
9. Figure 2: Crosses that mark the true value (zero in this particular case), are useful in order to visually check for correlations directly from the figure. Also, the authors state that "The constraints are somewhat tighter than those in the literature [13], as our MCMC exploration covers a smaller EMRIs’ parameter space". This is indeed probably one of the reasons to get smaller relative errors on the parameters, but other simplifying assumptions on the analysis (as speculated in previous points) could also contribute. Another reason could be the different version of instrumental noise. The one used here is quite outdated. In summary, I believe that more possible explanations should be added here and in the main text as a short discussion on the results.
B. Comments on the software
1. I believe that there are many benefits when (at least part of) the code is pip-installable. In my eyes, the most important element is the waveform implementation, which could be used as a direct plugin in other likelihoods and analysis pipelines. This could also expand the potential user-base of the software.
2. About the likelihood computation: Unless I am mistaken, the likelihood function is computed serially for each of the walkers. However, the emcee (or similar MCMC implementations) support vectorized likelihood outputs. This means that it is possible to build a likelihood function that has as input a matrix of parameter values [n_walkers x n_parameters], computes an array of residuals [n_walkers x n_datapoints] and then outputs a vector of likelihood [n_walkers, ] values, each corresponding to each walker. This allows for even higher efficiency with the GPU hardware, which is ideal for such vectorized operations.
3. From the code it is clear that the analysis is performed on 'noiseless' data, i.e. no noise is simulated. Then, the likelihood can be simplified to (d|h) - 0.5 (h|h) . E.g. see eq. (31) from Cañizares et al 2013.
We thank the Referee for pointing this out, however, we decided to retain the form of our likelihood in the code for practical reasons.
4. Unless I am mistaken, the noise is computed at each iteration, which is probably redundant, unless the noise is to be inferred from the data. A solution would be to precompute it and use it like the rest of the global constants. Another idea would be to transform the likelihood function into a likelihood class, which would compute the noise vector once and store it for use at each evaluation.
5. A very useful idea (but would require some extra work), is to make the code CPU/GPU agnostic. At the beginning of the script one can check if a GPU device is detected. If not, then the usual numpy library can be imported as import numpy as cp , and continue with the computations using the available CPUs. This adjustment would make the software more robust, and probably quite useful to the potential users with no access to GPUs.
Other changes in the code/manuscript:
1.We have improved the coding style at some parts of the code. 2.We have corrected typos in the documentation/comments within the code. 3.We have introduced the new command “run_code()” which executes the MCMC run once all parameters are defined, and we have adopted the manuscript to reflect new changes in the code. 4.We have improved the efficiency of the code by making even more variables global (pre-computed). 5.We made a slight change in the title of the paper, adding the word “Python”, infront of the word “code”. The new title now reads: “emri_mc: A GPU-based Python code for Bayesian inference of EMRI waveforms”. 6.We have improved the README file of our code providing detailed instructions for its manual installation. 7.Added a new comment in the draft (Section 5) to reflect some recent interesting works int he literature on machine learning methods.

---

## Editorial Decision

unknown